# Long-term causal effects via behavioral game theory

**Panagiotis (Panos) Toulis**
Econometrics & Statistics, Booth School
University of Chicago
Chicago, IL, 60637
panos.toulis@chicagobooth.edu

**David C. Parkes**
Department of Computer Science
Harvard University
Cambridge, MA, 02138
parkes@eecs.harvard.edu

## Abstract

Planned experiments are the gold standard in reliably comparing the causal effect of switching from a baseline policy to a new policy. One critical shortcoming of classical experimental methods, however, is that they typically do not take into account the dynamic nature of response to policy changes. For instance, in an experiment where we seek to understand the effects of a new ad pricing policy on auction revenue, agents may adapt their bidding in response to the experimental pricing changes. Thus, causal effects of the new pricing policy after such adaptation period, the *long-term causal effects*, are not captured by the classical methodology even though they clearly are more indicative of the value of the new policy. Here, we formalize a framework to define and estimate long-term causal effects of policy changes in multiagent economies. Central to our approach is behavioral game theory, which we leverage to formulate the ignorability assumptions that are necessary for causal inference. Under such assumptions we estimate long-term causal effects through a latent space approach, where a behavioral model of how agents act conditional on their latent behaviors is combined with a temporal model of how behaviors evolve over time.

## 1 Introduction

A multiagent economy is comprised of agents interacting under specific economic rules. A common problem of interest is to experimentally evaluate changes to such rules, also known as *treatments*, on an objective of interest. For example, an online ad auction platform is a multiagent economy, where one problem is to estimate the effect of raising the reserve price on the platform's revenue. Assessing causality of such effects is a challenging problem because there is a conceptual discrepancy between what needs to be estimated and what is available in the data, as illustrated in Figure 1.

What needs to be estimated is the *causal effect* of a policy change, which is defined as the difference between the objective value when the economy is treated, i.e., when *all* agents interact under the new rules, relative to when the same economy is in control, i.e., when *all* agents interact under the baseline rules. Such definition of causal effects is logically necessitated from the designer's task, which is to select either the treatment or the control policy based on their estimated revenues, and then apply such policy to all agents in the economy. The *long-term causal effect* is the causal effect defined after the system has stabilized, and is more representative of the value of policy changes in dynamical systems. Thus, in Figure 1 the long-term causal effect is the difference between the objective values at the top and bottom endpoints, marked as the "targets of inference".

What is available in the experimental data, however, typically comes from designs such as the so-called A/B test, where we randomly assign *some* agents to the treated economy (new rules B) and the others to the control economy (baseline rules A), and then compare the outcomes. In Figure 1 the experimental data are depicted as the solid time-series in the middle of the plot, marked as the "observed data".

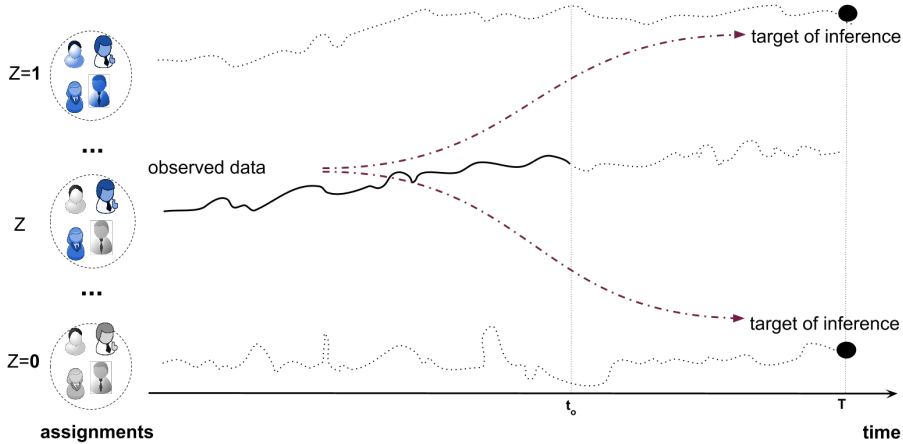

Figure 1: The two inferential tasks for causal inference in multiagent economies. First, infer agent actions across treatment assignments (y-axis), particularly, the assignment where all agents are in the treated economy (top assignment, $Z = 1$), and the assignment where all agents are in the control economy (bottom assignment, $Z = 0$). Second, infer across time, from $t_0$ (last observation time) to long-term $T$. What we seek in order to evaluate the causal effect of the new treatment is the difference between the objectives (e.g., revenue) at the two inferential target endpoints.

Therefore the challenge in estimating long-term causal effects is that we generally need to perform two inferential tasks simultaneously, namely,

  (i)  infer outcomes across possible experimental policy assignments (y-axis in Figure 1), and

  (ii)  infer long-term outcomes from short-term experimental data (x-axis in Figure 1).

The first task is commonly known as the "fundamental problem of causal inference" [14, 19] because it underscores the impossibility of observing in the same experiment the outcomes for *both* policy assignments that define the causal effect; i.e., that we cannot observe in the same experiment both the outcomes when all agents are treated and the outcomes when all agents are in control, the assignments of which are denoted by $Z = \mathbf{1}$ and $Z = \mathbf{0}$, respectively, in Figure 1. In fact the role of experimental design, as conceived by R.A. Fisher [8], is exactly to quantify the uncertainty about such causal effects that cannot be observed due to the aforementioned fundamental problem, by using standard errors that can be observed in a carefully designed experiment.

The second task, however, is unique to causal inference in dynamical systems, such as the multiagent economies that we study in this paper, and has received limited attention so far. Here, we argue that it is crucial to study long-term causal effects, i.e., effects measured after the system has stabilized, because such effects are more representative of the value of policy changes. If our analysis focused only on the observed data part depicted in Figure 1, then policy evaluation would reflect transient effects that might differ substantially from the long-term effects. For instance, raising the reserve price in an auction might increase revenue in the short-term but as agents adapt their bids, or switch to another platform altogether, the long-term effect could be a net decrease in revenue [13].

## 1.1   Related work and our contributions

There have been several important projects related to causal inference in multiagent economies. For instance, Ostrovsky and Schwartz [16] evaluated the effects of an increase in the reserve price of Yahoo! ad auctions on revenue. Auctions were randomly assigned to an increased reserve price treatment, and the effect was estimated using difference-in-differences (DID), which is a popular econometric method [6, 7, 16]. In relation to Figure 1, DID extrapolates across assignments (y-axis) and across time (x-axis) by making a strong additivity assumption [1, 3, Section 5.2], specifically, by assuming that the dependence of revenue on reserve price and time is additive.

In a structural approach, Athey et.al. [4] studied the effects of auction format (ascending versus sealed bid) on competition for timber tracts. In relation to Figure 1, their approach extrapolates

across assignments by assuming that agent individual valuations for tracts are independent of the treatment assignment, and extrapolates across time by assuming that the observed agent bids are already in equilibrium. Similar approaches are followed in econometrics for estimation of general equilibrium effects [11, 12].

In a causal graph approach [17] Bottou et.al. [5] studied effects of changes in the algorithm that scores Bing ads on the ad platform's revenue. In relation to Figure 1, their approach is non-experimental and extrapolates across assignments and across time by assuming a directed acyclic graph (DAG) as the correct data model, which is also assumed to be stable with respect to treatment assignment, and by estimating counterfactuals through the fitted model.

Our work is different from prior work because it takes into account the short-term aspect of experimental data to evaluate long-term causal effects, which is the key conceptual and practical challenge that arises in empirical applications. In contrast, classical econometric methods, such as DID, assume strong linear trends from short-term to long-term, whereas structural approaches typically assume that the experimental data are already long-term as they are observed in equilibrium. We refer the reader to Sections 2 and 3 of the supplement for more detailed comparisons.

In summary, our key contribution is that we develop a formal framework that (i) articulates the distinction between short-term and long-term causal effects, (ii) leverages behavioral game-theoretic models for causal analysis of multiagent economies, and (iii) explicates theory that enables valid inference of long-term causal effects.

## 2 Definitions

Consider a set of agents $\mathcal{I}$ and a set of actions $\mathcal{A}$, indexed by $i$ and $a$, respectively. The experiment designer wants to run an experiment to evaluate a new policy against the baseline policy relative to an objective. In the experiment each agent is assigned to one policy, and the experimenter observes how agents act over time. Formally, let $Z = (Z_i)$ be the $|\mathcal{I}| \times 1$ assignment vector where $Z_i = 1$ denotes that agent $i$ is assigned to the new policy, and $Z_i = 0$ denotes that $i$ is assigned to the baseline policy; as a shorthand, $Z = \mathbf{1}$ denotes that all agents are assigned to the new policy, and $Z = \mathbf{0}$ denotes that all agents are assigned to the baseline policy, where $\mathbf{1}, \mathbf{0}$ generally denote an appropriately-sized vector of ones and zeroes, respectively. In the simplest case, the experiment is an A/B test, where $Z$ is uniformly random on $\{0,1\}^{|\mathcal{I}|}$ subject to $\sum_i Z_i = |\mathcal{I}|/2$.

After the initial assignment $Z$ agents play actions at discrete time points from $t = 0$ to $t = t_0$. Let $A_i(t; Z) \in \mathcal{A}$ be the random variable that denotes the action of agent $i$ at time $t$ under assignment $Z$. The *population action* $\alpha_j(t; Z) \in \Delta^{|\mathcal{A}|}$, where $\Delta^p$ denotes the $p$-dimensional simplex, is the frequency of actions at time $t$ under assignment $Z$ of agents that were assigned to game $j$; for example, assuming two actions $\mathcal{A} = \{a_1, a_2\}$, then $\alpha_1(0; Z) = [0.2, 0.8]$ denotes that, under assignment $Z$, 20% of agents assigned to the new policy play action $a_1$ at $t = 0$, while the rest play $a_2$. We assume that the objective value for the experimenter depends on the population action, in a similar way that, say, auction revenue depends on agents' aggregate bidding. The objective value in policy $j$ at time $t$ under assignment $Z$ is denoted by $R(\alpha_j(t; Z))$, where $R : \Delta^{|\mathcal{A}|} \to \mathbb{R}$. For instance, suppose in the previous example that $a_1$ and $a_2$ produce revenue \$10 and $-\$2$, respectively, each time they are played, then $R$ is linear and $R([.2, .8]) = 0.2 \cdot \$10 - 0.8 \cdot \$2 = \$0.4$.

**Definition 1** *The average causal effect on objective $R$ at time $t$ of the new policy relative to the baseline is denoted by* $\mathrm{CE}(t)$ *and is defined as*
$$\mathrm{CE}(t) = \mathbb{E}\left(R(\alpha_1(t; \mathbf{1})) - R(\alpha_0(t; \mathbf{0}))\right). \tag{1}$$

Suppose that $(t_0, T]$ is the time interval required for the economy to adapt to the experimental conditions. The exact definition of $T$ is important but we defer this discussion for Section 3.1. The designer concludes that the new policy is better than the baseline if $\mathrm{CE}(T) > 0$. Thus, $\mathrm{CE}(T)$ is the *long-term average causal effect* and is a function of two objective values, $R(\alpha_1(T; \mathbf{1}))$ and $R(\alpha_0(T; \mathbf{0}))$, which correspond to the two inferential target endpoints in Figure 1. Neither value is observed in the experiment because agents are randomly split between policies, and their actions are observed only for the short-term period $[0, t_0]$. Thus we need to (i) extrapolate across assignments by pivoting from the observed assignment to the counterfactuals $Z = \mathbf{1}$ and $Z = \mathbf{0}$; (ii) extrapolate across time from the short-term data $[0, t_0]$ to the long-term $t = T$. We perform these two extrapolations based on a latent space approach, which is described next.

## 2.1 Behavioral and temporal models

We assume a latent behavioral model of how agents select actions, inspired by models from behavioral game theory. The behavioral model is used to predict agent actions conditional on agent behaviors, and is combined with a temporal model to predict behaviors in the long-term. The two models are ultimately used to estimate agent actions in the long-term, and thus estimate long-term causal effects. As the choice of the latent space is not unique, in Section 3.1 we discuss why we chose to use behavioral models from game theory.

Let $B_i(t; Z)$ denote the behavior that agent $i$ adopts at time $t$ under experimental assignment $Z$. The following assumption puts a constraints on the space of possible behaviors that agents can adopt, which will simplify the subsequent analysis.

**Assumption 1 (Finite set of possible behaviors)** *There is a fixed and finite set of behaviors $\mathcal{B}$ such that for every time $t$, assignment $Z$ and agent $i$, it holds that $B_i(t; Z) \in \mathcal{B}$; i.e., every agent can only adopt a behavior from $\mathcal{B}$.*

**Definition 2 (Behavioral model)** *The behavioral model for policy $j$ defined by set $\mathcal{B}$ of behaviors is the collection of probabilities*

$$P(A_i(t; Z) = a | B_i(t; Z) = b, G_j), \tag{2}$$

*for every action $a \in \mathcal{A}$ and every behavior $b \in \mathcal{B}$, where $G_j$ denotes the characteristics of policy $j$.*

As an example, a non-sophisticated behavior $b_0$ could imply that $P(A_i(t; Z) = a | b_0, G_j) = 1/|\mathcal{A}|$, i.e., that the agent adopting $b_0$ simply plays actions at random. Conditioning on policy $j$ in Definition 2 allows an agent to choose its actions based on expected payoffs, which depend on the policy characteristics. For instance, in the application of Section 4 we consider a behavioral model where an agent picks actions in a two-person game according to expected payoffs calculated from the game-specific payoff matrix—in that case $G_j$ is simply the payoff matrix of game $j$.

The *population behavior* $\beta_j(t; Z) \in \Delta^{|\mathcal{B}|}$ denotes the frequency at time $t$ under assignment $Z$ of the adopted behaviors of agents assigned to policy $j$. Let $\mathcal{F}_t$ denote the entire history of population behaviors in the experiment up to time $t$. A temporal model of behaviors is defined as follows.

**Definition 3 (Temporal model)** *For an experimental assignment $Z$ a temporal model for policy $j$ is a collection of parameters $\phi_j(Z), \psi_j(Z)$, and densities $(\pi, f)$, such that for all $t$,*

$$\beta_j(0; Z) \sim \pi(\cdot; \phi_j(Z)),$$
$$\beta_j(t; Z) | \ \mathcal{F}_{t-1}, G_j \sim f(\cdot | \psi_j(Z), \mathcal{F}_{t-1}). \tag{3}$$

A temporal model defines the distribution of population behavior as a time-series with a Markovian structure. As defined, the temporal model imposes the restriction that the prior $\pi$ of population behavior at $t = 0$ and the density $f$ of behavioral evolution are both independent of treatment assignment $Z$. In other words, regardless of how agents are assigned to games, the population behavior in the game will evolve according to a fixed model described by $f$ and $\pi$. The model parameters $\phi, \psi$ may still depend on the treatment assignment $Z$.

## 3 Estimation of long-term causal effects

Here we develop the assumptions that are necessary for inference of long-term causal effects.

**Assumption 2 (Stability of initial behaviors)** *Let $\rho_Z = \sum_{i \in \mathcal{I}} Z_i / |\mathcal{I}|$ be the proportion of agents assigned to the new policy under assignment $Z$. Then, for every possible assignment $Z$,*

$$\rho_Z \beta_1(0; Z) + (1 - \rho_Z)\beta_0(0; Z) = \beta^{(0)}, \tag{4}$$

*where $\beta^{(0)}$ is a fixed population behavior invariant to $Z$.*

**Assumption 3 (Behavioral ignorability)** *The assignment is independent of population behavior at time $t$, conditional on policy and behavioral history up to $t$; i.e., for every $t > 0$ and policy $j$,*

$$Z \perp\!\!\!\perp \beta_j(t; Z) \mid \mathcal{F}_{t-1}, G_j.$$

*Remarks.* Assumption 2 implies that the agents do not anticipate the assignment $Z$ as they "have made up their minds" to adopt a population behavior $\beta^{(0)}$ before the experiment. Quantities such as that in Eq. (4) are crucial in causal inference because they can be used as a pivot for extrapolation across assignments. Assumption 3 states that the treatment assignment does not add information about the population behavior at time $t$, if we already know the full behavioral history of up to $t$, and the policy which agents are assigned to; hence, the treatment assignment is conditionally *ignorable*. This ignorability assumption precludes, for instance, an agent adopting a different behavior depending on whether it was assigned with friends or foes in the experiment.

Algorithm 1 is the main methodological contribution of this paper. It is a Bayesian procedure as it puts priors on parameters $\phi, \psi$ of the temporal model, and then marginalizes these parameters out.

---

**Algorithm 1** Estimation of long-term causal effects
**Input:** $Z, T, \mathcal{A}, \mathcal{B}, G_1, G_0, \mathcal{D}_1 = \{a_1(t; Z) : t = 0, \ldots, t_0\}, \mathcal{D}_0 = \{a_0(t; Z) : t = 0, \ldots, t_0\}$.
**Output:** Estimate of long-term causal effect $\mathrm{CE}(T)$ in Eq. (1).

1: By Assumption 3, define $\phi_j \equiv \phi_j(Z)$, $\psi_j \equiv \psi_j(Z)$.
2: Set $\mu_1 \leftarrow \mathbf{0}$ and $\mu_0 \leftarrow \mathbf{0}$, both of size $|\mathcal{A}|$; set $\nu_0 = \nu_1 = 0$.
3: **for** $iter = 1, 2, \ldots$ **do**
4:      For $j = 0, 1$, sample $\phi_j, \psi_j$ from prior, and sample $\beta_j(0; Z)$ conditional on $\phi_j$.
5:      Calculate $\beta^{(0)} = \rho_Z \beta_1(0; Z) + (1 - \rho_Z)\beta_0(0; Z)$.
6:      **for** $j = 0, 1$ **do**
7:          Set $\beta_j(0; j\mathbf{1}) = \beta^{(0)}$.
8:          Sample $B_j = \{\beta_j(t; j\mathbf{1}) : t = 0, \ldots, T\}$ given $\psi_j$ and $\beta_j(0, j\mathbf{1})$.     *# temporal model*
9:          Sample $\alpha_j(T; j\mathbf{1})$ conditional on $\beta_j(T; j\mathbf{1})$.     *# behavioral model*
10:         Set $\mu_j \leftarrow \mu_j + P(\mathcal{D}_j | B_j, G_j) \cdot R(\alpha_j(T; j\mathbf{1}))$.
11:         Set $\nu_j \leftarrow \nu_j + P(\mathcal{D}_j | B_j, G_j)$.
12:      **end for**
13: **end for**
14: Return estimate $\widehat{\mathrm{CE}}(T) = \mu_1/\nu_1 - \mu_0/\nu_0$.

---

**Theorem 1 (Estimation of long-term causal effects)** *Suppose that behaviors evolve according to a known temporal model, and actions are distributed conditionally on behaviors according to a known behavioral model. Suppose that Assumptions 1, 2 and 3 hold for such models. Then, for every policy $j \in \{0, 1\}$ as the iterations of Algorithm 1 increase, $\mu_j/\nu_j \to \mathbb{E}\left(R(\alpha_j(T; j\mathbf{1})) | \mathcal{D}_j\right)$. The output $\widehat{\mathrm{CE}}(T)$ of Algorithm 1 asymptotically estimates the long-term causal effect, i.e.,*

$$\mathbb{E}(\widehat{\mathrm{CE}}(T)) = \mathbb{E}\left(R(\alpha_1(T; \mathbf{1})) - R(\alpha_0(T; \mathbf{0}))\right) \equiv \mathrm{CE}(T).$$

*Remarks.* Theorem 1 shows that $\widehat{\mathrm{CE}}(T)$ consistently estimates the long-term causal effect in Eq. (1). We note that it is also possible to derive the variance of this estimator with respect to the randomization distribution of assignment $Z$. To do so we first create a set of assignments $\mathcal{Z}$ by repeatedly sampling $Z$ according to the experimental design. Then we adapt Algorithm 1 so that (i) Step 4 is removed; (ii) in Step 5, $\beta^{(0)}$ is sampled from its posterior distribution conditional on observed data, which can be obtained from the original Algorithm 1. The empirical variance of the outputs over $\mathcal{Z}$ from the adapted algorithm estimates the variance of the output $\widehat{\mathrm{CE}}(T)$ of the original algorithm. We leave the full characterization of this variance estimation procedure for future work.

### 3.1 Discussion

Methodologically, our approach is aligned with the idea that for long-term causal effects we need a model for outcomes that leverages structural information pertaining to how outcomes are generated and how they evolve. In our application such structural information is the microeconomic information that dictates what agent behaviors are successful in a given policy and how these behaviors evolve over time.

In particular, Step 1 in the algorithm relies on Assumptions 2 and 3 to infer that model parameters, $\phi_j, \psi_j$ are stable with respect to treatment assignment. Step 5 of the algorithm is the key estimation pivot, which uses Assumption 2 to extrapolate from the experimental assignment $Z$ to the counterfactual assignments $Z = \mathbf{1}$ and $Z = \mathbf{0}$, as required in our problem. Having pivoted to such

counterfactual assignment, it is then possible to use the temporal model parameters $\psi_j$, which are unaffected by the pivot under Assumption 3, to sample population behaviors up to long-term $T$, and subsequently sample agent actions at $T$ (Steps 8 and 9).

Thus, a lot of burden is placed on the behavioral game-theoretic model to predict agent actions, and the accuracy of such models is still not settled [10]. However, it does not seem necessary that such prediction is completely accurate, but rather that the behavioral models can pull relevant information from data that would otherwise be inaccessible without game theory, thereby improving over classical methods. A formal assessment of such improvement, e.g., using information theory, is open for future work. An empirical assessment can be supported by the extensive literature in behavioral game theory [20, 15], which has been successful in predicting human actions in real-world experiments [22].

Another limitation of our approach is Assumption 1, which posits that there is a finite set of pre-defined behaviors. A nonparametric approach where behaviors are estimated on-the-fly might do better. In addition, the long-term horizon, $T$, also needs to be defined *a priori*. We should be careful how $T$ interferes with the temporal model since such a model implies a time $T'$ at which population behavior reaches stationarity. Thus if $T' \leq T$ we implicitly assume that the long-term causal effect of interest pertains to a stationary regime (e.g., Nash equilibrium), but if $T' > T$ we assume that the effect pertains to a transient regime, and therefore the policy evaluation might be misguided.

# 4    Application: Long-term causal effects from a behavioral experiment

In this section, we apply our methodology to experimental data from Rapoport and Boebel [18], as reported by McKelvey and Palfrey [15]. The experiment consisted of a series of zero-sum two-agent games, and aimed at examining the hypothesis that human players play according to minimax solutions of the game, the so-called minimax hypothesis initially suggested by von Neumann and Morgenstern [21]. Here we repurpose the data in a slightly artificial way, including how we construct the designer's objective. This enables a suitable demonstration of our approach.

Each game in the experiment was a simultaneous-move game with five discrete actions for the row player and five actions for the column player. The structure of the payoff matrix, given in the supplement in Table 1, is parametrized by two values, namely $W$ and $L$; the experiment used two different versions of payoff matrices, corresponding to payments by the row agent to the column agent when the row agent *won* ($W$), or *lost* ($L$): modulo a scaling factor, Rapoport and Boebel [18] used $(W, L) = (\$10, -\$6)$ for game 0 and $(W, L) = (\$15, -\$1)$ for game 1.

Forty agents, $\mathcal{I} = \{1, 2, \ldots, 40\}$, were randomized to one game design (20 agents per game), and each agent played once as row and once as column, matched against two different agents. Every match-up between a pair of agents lasted for two periods of 60 rounds, with each round consisting of a selection of an action from each agent and a payment. Thus, each agent played for four periods and 240 rounds in total. If $Z$ is the entire assignment vector of length 40, $Z_i = 1$ means that agent $i$ was assigned to game 1 with payoff matrix $(W, L) = (\$15, -\$1)$ and $Z_i = 0$ means that $i$ was assigned to game 0 with payoff matrix $(W, L) = (\$10, -\$6)$.

In adapting the data, we take advantage of the randomization in the experiment, and ask a question in regard to long-term causal effects. In particular, assuming that agents pay a fee for each action taken, which accounts for the revenue of the game, we ask the following question:

*"What is the long-term causal effect on revenue if we switch from payoffs $(W, L) = (\$10, -\$6)$ of game 0 to payoffs $(W, L) = (\$15, -\$1)$ of game 1?".*

The games induced by the two aforementioned payoff matrices represent the two different policies we wish to compare. To evaluate our method, we consider the last period as long-term, and hold out data from this period. We define the causal estimand in Eq. (1) as

$$\mathrm{CE} = c^{\mathsf{T}}(\alpha_1(T; \mathbf{1}) - \alpha_0(T; \mathbf{0})), \tag{5}$$

where $T = 3$ and $c$ is a vector of coefficients. The interpretation is that, given an element $c_a$ of $c$, the agent playing action $a$ is assumed to pay a constant fee $c_a$. To check the robustness of our method we test Algorithm 1 over multiple values of $c$.

## 4.1 Implementation of Algorithm 1 and results

Here we demonstrate how Algorithm 1 can be applied to estimate the long-term causal effect in Eq. (5) on the Rapoport & Boebel dataset. To this end we clarify Algorithm 1 step by step, and give more details in the supplement.

**Step 1: Model parameters.** For simplicity we assume that the models in the two games share common parameters, and thus $(\phi_1, \psi_1, \lambda_1) = (\phi_0, \psi_0, \lambda_0) \equiv (\phi, \psi, \lambda)$, where $\lambda$ are the parameters of the behavioral model to be described in Step 8. Having common parameters also acts as regularization and thus helps estimation.

**Step 4: Sampling parameters and initial behaviors** As explained later we assume that there are 3 different behaviors and thus $\phi, \psi, \lambda$ are vectors with 3 components. Let $x \sim U(m, M)$ denote that every component of $x$ is uniform on $(m, M)$, independently. We choose diffuse priors for our parameters, specifically, $\phi \sim \mathrm{U}(0, 10)$, $\psi \sim \mathrm{U}(-5, 5)$, and $\lambda \sim \mathrm{U}(-10, 10)$. Given $\phi$ we sample the initial behaviors as Dirichlet, i.e., $\beta_1(0; Z) \sim \mathrm{Dir}(\phi)$ and $\beta_0(0; Z) \sim \mathrm{Dir}(\phi)$, independently.

**Steps 5 & 7: Pivot to counterfactuals.** Since we have a completely randomized experiment (A/B test) it holds that $\rho_Z = 0.5$ and therefore $\beta^{(0)} = 0.5(\beta_1(0; Z) + \beta_0(0; Z))$. Now we can pivot to the counterfactual population behaviors under $Z = \mathbf{1}$ and $Z = \mathbf{0}$ by setting $\beta_1(0; \mathbf{1}) = \beta_0(0; \mathbf{0}) = \beta^{(0)}$.

**Step 8: Sample counterfactual behavioral history.** As the temporal model, we adopt the *lag-one vector autoregressive model*, also known as VAR(1). We transform[1] the population behavior into a new variable $w_t = \mathrm{logit}(\beta_1(t; \mathbf{1})) \in \mathbb{R}^2$ (also do so for $\beta_0(t; \mathbf{0})$). Such transformation with a unique inverse is necessary because population behaviors are constrained on the simplex, and thus form so-called compositional data [2, 9]. The VAR(1) model implies that

$$w_t = \psi[1]\mathbf{1} + \psi[2]w_{t-1} + \psi[3]\epsilon_t, \tag{6}$$

where $\psi[k]$ is the $k$th component of $\psi$ and $\epsilon_t \sim \mathcal{N}(0, I)$ is i.i.d. standard bivariate normal. Eq. (6) is used to sample the behavioral history, $B_j$, in Step 8 of Algorithm 1.

**Step 9: Behavioral model.** For the behavioral model, we adopt the *quantal p-response* ($\mathrm{QL}_p$) model [20], which has been successful in predicting human actions in real-world experiments [22]. We choose $p = 3$ behaviors, namely $\mathcal{B} = \{b_0, b_1, b_2\}$ of increased sophistication parametrized by $\lambda = (\lambda[1], \lambda[2], \lambda[3]) \in \mathbb{R}^3$. Let $G_j$ denote the $5 \times 5$ payoff matrix of game $j$ and let the term *strategy* denote a distribution over all actions. An agent with behavior $b_0$ plays the uniform strategy,

$$P(A_i(t; Z) = a | B_i(t; Z) = b_0, G_j) = 1/5.$$

An agent of level-1 (row player) assumes to be playing only against level-0 agents and thus expects per-action profit $u_1 = (1/5)G_j\mathbf{1}$ (for column player we use the transpose of $G_j$). The level-1 agent will then play a strategy proportional to $e^{\lambda[1]u_1}$, where $e^x$ for vector $x$ denotes the element-wise exponentiation, $e^x = (e^{x[k]})$. The precision parameter $\lambda[1]$ determines how much an agent insists on maximizing expected utility; for example, if $\lambda[1] = \infty$, the agent plays the action with maximum expected payoff (best response); if $\lambda[1] = 0$, the agent acts as a level-0 agent. An agent of level-2 (row player) assumes to be playing only against level-1 agents with precision $\lambda[2]$ and therefore expects to face strategy proportional to $e^{\lambda[2]u_1}$. Thus its expected per-action profit is $u_2 \propto G_j e^{\lambda[2]u_1}$, and plays strategy $\propto e^{\lambda[3]u_2}$.

Given $G_j$ and $\lambda$ we calculate a $5 \times 3$ matrix $Q_j$ where the $k$th column is the strategy played by an agent with behavior $b_{k-1}$. The expected population action is therefore $\bar{\alpha}_j(t; Z) = Q_j\beta_j(t; Z)$. The population action $\alpha_j(t; Z)$ is distributed as a normalized multinomial random variable with expectation $\bar{\alpha}_j(t; Z)$, and so $P(\alpha_j(t; \mathbf{1}) | \beta_j(t; \mathbf{1}), G_j) = \mathrm{Multi}(|\mathcal{I}| \cdot \alpha_j(t; \mathbf{1}); \bar{\alpha}_j(t; \mathbf{1}))$, where $\mathrm{Multi}(n; p)$ is the multinomial density of observations $n = (n_1, \ldots, n_K)$ with probabilities $p = (p_1, \ldots, p_K)$. Hence, the full likelihood for observed actions in game $j$ in Steps 10 and 11 of Algorithm 1 is given by the product

$$P(\mathcal{D}_j | B_j, G_j) = \prod_{t=0}^{T-1} \mathrm{Multi}(|\mathcal{I}| \cdot \alpha_j(t; j\mathbf{1}); \bar{\alpha}_j(t; j\mathbf{1})).$$

Running Algorithm 1 on the Rapoport and Boebel dataset yields the estimates shown in Figure 2, for 25 different fee vectors $c$, where each component $c_a$ is sampled uniformly at random from $(0, 1)$.

Figure 2: Estimates of long-term effects of different methods corresponding to 25 random objective coefficients $c$ in Eq. (5). For estimates of our method we ran Algorithm 1 for 100 iterations.

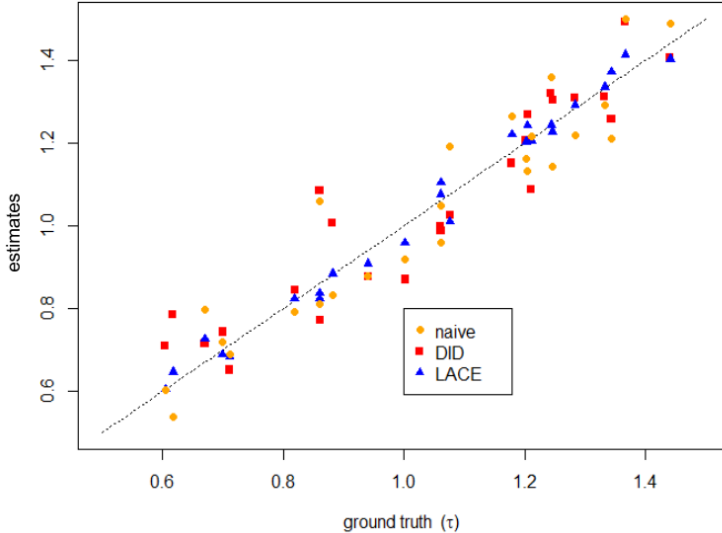

We also test difference-in-differences (DID), which estimates the causal effect through

$$\hat{\tau}^{did} = [R(\alpha_1(2; Z)) - R(\alpha_1(0; Z))] - [R(\alpha_0(2; Z)) - R(\alpha_0(0; Z))],$$

and a naive method ("naive" in the plot), which ignores the dynamical aspect and estimates the long-term causal effect as $\hat{\tau}^{nai} = [R(\alpha_1(2; Z)) - R(\alpha_0(2; Z))]$. Our estimates ("LACE" in the plot) are closer to the truth (mse $= 0.045$) than the estimates from the naive method (mse $= 0.185$) and from DID (mse $= 0.361$). This illustrates that our method can pull game-theoretic information from the data for long-term causal inference, whereas the other methods cannot.

## 5   Conclusion

One critical shortcoming of statistical methods of causal inference is that they typically do not assess the long-term effect of policy changes. Here we combined causal inference and game theory to build a framework for estimation of such long-term effects in multiagent economies. Central to our approach is behavioral game theory, which provides a natural latent space model of how agents act and how their actions evolve over time. Such models enable to predict how agents would act under various policy assignments and at various time points, which is key for valid causal inference. Working on a real-world dataset [18] we showed how our framework can be applied to estimate the long-term effect of changing the payoff structure of a normal-form game.

Our framework could be extended in future work by incorporating learning (e.g., fictitious play, bandits, no-regret learning) to better model the dynamic response of multiagent systems to policy changes. Another interesting extension would be to use our framework for optimal design of experiments in such systems, which needs to account for heterogeneity in agent learning capabilities and for intrinsic dynamical properties of the systems' responses to experimental treatments.

## Acknowledgements

The authors wish to thank Leon Bottou, the organizers and participants of CODE@MIT'15, GAMES'16, the Workshop on Algorithmic Game Theory and Data Science (EC'15), and the anonymous NIPS reviewers for their valuable feedback. Panos Toulis has been supported in part by the 2012 Google US/Canada Fellowship in Statistics. David C. Parkes was supported in part by NSF grant CCF-1301976 and the SEAS TomKat fund.

## Footnotes

[1] $y = \mathrm{logit}(x)$ is defined as the function $\Delta^m \to \mathbb{R}^{m-1}$, $y[i] = \log(x[i+1]/x[1])$, where $x[1] \neq 0$ wlog.

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
