[Supplementary Material · LongTermCausalEffects_final_3 (supplement).pdf]

# Long-term causal effects via behavioral game theory
*Supplementary material for NIPS 2016*

Panos Toulis (U. Chicago) and David C. Parkes (Harvard)

## 1 Proof of Theorem 1

**Definition 1.** *The average causal effect on objective $R$ at time $t$ of the new policy relative to the baseline is denoted by $CE(t)$ and is defined as*

$$CE(t) = \mathbb{E}\left(R(\alpha_1(t; \mathbf{1})) - R(\alpha_0(t; \mathbf{0}))\right). \tag{1}$$

**Assumption 1** (Finite set of possible behaviors). *There is a fixed and finite set of behaviors $\mathcal{B}$ such that for every time $t$, assignment $Z$ and agent $i$, it holds that $B_i(t; Z) \in \mathcal{B}$; i.e., every agent can only adopt a behavior from $\mathcal{B}$.*

**Assumption 2** (Stability of initial behaviors). *Let $\rho_Z = \sum_{i \in \mathcal{I}} Z_i / |\mathcal{I}|$ be the proportion of agents assigned to the new policy under assignment $Z$. Then there exists a fixed population behavior $\beta^{(0)}$ such that for every possible assignment $Z$,*

$$\rho_Z \beta_1(0; Z) + (1 - \rho_Z)\beta_0(0; Z) = \beta^{(0)}. \tag{2}$$

**Assumption 3** (Behavioral ignorability). *The assignment is independent of population behavior at time $t$, conditional on policy and behavioral history up to $t$; i.e., for every $t > 0$ and policy $j$,*

$$Z \perp\!\!\!\perp \beta_j(t; Z) \mid \mathcal{F}_{t-1}, G_j.$$

---

**Algorithm 1** Estimation of long-term causal effects
**Input:** $Z, T, \mathcal{A}, \mathcal{B}, G_1, G_0, \mathcal{D}_1 = \{a_1(t; Z) : t = 0, \ldots, t_0\}, \mathcal{D}_0 = \{a_0(t; Z) : t = 0, \ldots, t_0\}$.
**Output:** Estimate of long-term causal effect $CE(T)$ in Eq. (1).

---
1: By Assumption 3, define $\phi_j \equiv \phi_j(Z)$, $\psi_j \equiv \psi_j(Z)$.
2: Set $\mu_1 \leftarrow \mathbf{0}$ and $\mu_0 \leftarrow \mathbf{0}$, both of size $|\mathcal{A}|$; set $\nu_0 = \nu_1 = 0$.
3: **for** $iter = 1, 2, \ldots$ **do**
4:     For $j = 0, 1$, sample $\phi_j, \psi_j$ from prior, and sample $\beta_j(0; Z)$ conditional on $\phi_j$.
5:     Calculate $\beta^{(0)} = \rho_Z \beta_1(0; Z) + (1 - \rho_Z)\beta_0(0; Z)$.
6:     **for** $j = 0, 1$ **do**
7:         Set $\beta_j(0; j\mathbf{1}) = \beta^{(0)}$.
8:         Sample $B_j = \{\beta_j(t; j\mathbf{1}) : t = 0, \ldots, T\}$ given $\psi_j$ and $\beta_j(0, j\mathbf{1})$.     # *temporal model*
9:         Sample $\alpha_j(T; j\mathbf{1})$ conditional on $\beta_j(T; j\mathbf{1})$.     # *behavioral model*
10:        Set $\mu_j \leftarrow \mu_j + P(\mathcal{D}_j | B_j, G_j) \cdot R(\alpha_j(T; j\mathbf{1}))$.
11:        Set $\nu_j \leftarrow \nu_j + P(\mathcal{D}_j | B_j, G_j)$.
12:     **end for**
13: **end for**
14: Return estimate $\widehat{CE}(T) = \mu_1/\nu_1 - \mu_0/\nu_0$.

---

**Theorem 1** (Estimation of long-term causal effects). *Suppose that behaviors evolve according to a known temporal model, and actions are distributed conditionally on behaviors according to a known behavioral model. Suppose that Assumptions 1, 2 and 3 hold for such models. Then, for every policy $j \in \{0, 1\}$ as the iterations of Algorithm 1 increase, $\mu_j / \nu_j \to \mathbb{E}\left(R(\alpha_j(T; j\mathbf{1}))|\mathcal{D}_j\right)$. The output $\widehat{CE}(T)$ of Algorithm 1 asymptotically estimates the long-term causal effect, i.e.,*

$$\mathbb{E}(\widehat{CE}(T)) = \mathbb{E}\left(R(\alpha_1(T; \mathbf{1})) - R(\alpha_0(T; \mathbf{0}))\right) \equiv CE(T).$$

*Proof.* Fix a policy $j$ in Algorithm 1 and drop the subscript $j$ in the notation of the algorithm. Therefore we can write:

$$\omega \equiv (\phi_j, \psi_j, B_j)$$
$$\alpha \equiv \alpha_j(T; j\mathbf{1})$$
$$P(\mathcal{D}|\omega) \equiv P(\mathcal{D}_j|B_j, G_j). \tag{3}$$

The way Algorithm 1 is defined, as the iterations increase the variable $\mu$ is estimating

$$\lim \mu = \int R(\alpha)P(\mathcal{D}|\omega)p(\alpha, \omega)d\omega d\alpha.$$

We now rewrite this integral as follows.

$$\lim \mu = \int R(\alpha)P(\mathcal{D}|\omega)p(\alpha, \omega)d\omega d\alpha = \int R(\alpha)P(\mathcal{D}|\alpha, \omega)p(\alpha, \omega)d\omega d\alpha \quad [\text{because } p(\mathcal{D}|\alpha, \omega) = P(\mathcal{D}|\omega)]$$

$$= \int R(\alpha)P(\alpha, \omega|\mathcal{D})P(\mathcal{D})d\omega d\alpha \quad [\text{by Bayes theorem}]$$

$$= P(\mathcal{D}) \int R(\alpha)P(\alpha|\mathcal{D})d\alpha \quad [\omega \text{ is marginalized out}]$$

$$= P(\mathcal{D})\mathbb{E}\left(R(\alpha)|\mathcal{D}\right). \tag{4}$$

The first equation, $p(\mathcal{D}|\alpha, \omega) = P(\mathcal{D}|\omega)$, holds by definition of the behavioral model: the history of latent behaviors is sufficient for the likelihood of observed actions. Another way to phrase this is that conditional on latent behavior the observed action is independent from any other variable.

Similarly, as the iterations increase the variable $\nu$ is estimating

$$\lim \nu = \int P(\mathcal{D}|\omega)p(\alpha, \omega)d\omega d\alpha.$$

We now rewrite this integral as follows.

$$\lim \nu = \int P(\mathcal{D}|\omega)p(\alpha, \omega)d\omega d\alpha = \int P(\mathcal{D}|\alpha, \omega)p(\alpha, \omega)d\omega d\alpha \quad [\text{because } p(\mathcal{D}|\alpha, \omega) = P(\mathcal{D}|\omega)]$$

$$= \int P(\alpha, \omega|\mathcal{D})P(\mathcal{D})d\omega d\alpha \quad [\text{by Bayes theorem}]$$

$$= P(\mathcal{D}) \int P(\alpha|\mathcal{D})d\alpha$$

$$= P(\mathcal{D}). \tag{5}$$

By the continuous mapping theorem we conclude that

$$\lim \mu/\nu \to \mathbb{E}\left(R(\alpha)|\mathcal{D}\right).$$

Thus $\mathbb{E}\left(\lim \mu_1/\nu_1\right) = \mathbb{E}\left(R(\alpha_1(T; \mathbf{1}))\right)$ and $\mathbb{E}\left(\lim \mu_0/\nu_0\right) = \mathbb{E}\left(R(\alpha_0(T; \mathbf{0}))\right)$ and so

$$\mathbb{E}\left(\lim \mu_1/\nu_1\right) - \mathbb{E}\left(\lim \mu_0/\nu_0\right) \to \mathbb{E}\left(R(\alpha_1(T; \mathbf{1}))\right) - \mathbb{E}\left(R(\alpha_0(T; \mathbf{0}))\right),$$

i.e., Algorithm 1 consistently estimates the long-term causal effect. $\qquad\square$

## 2 Connection of assumptions to policy invariance

Assumption 3 in our framework is related to *policy invariance* assumptions in econometrics of policy effects [14, 13]. Intuitively, policy invariance posits that given the *choice* of policy by an agent, the initial process that resulted in this choice does not affect the outcome. For example, given that an individual chooses to participate in a tax benefit program, the way the individual was assigned to the program (e.g., lottery, recommendation, or point of a gun) does not alter the outcome that will be observed for that individual. Our assumption is different because we have a temporal evolution of population behavior and there is no free choice of an agent about the assignment, since we assume a randomized experiment. But our assumption shares the essential aspect of conditional ignorability of assignment that is crucial in causal inference.

## 3 Discussion of related methods

Consider the estimand for the Rapoport-Boebel experiment [18]:

$$\tau = c^{\mathsf{T}}(\alpha_1(T; \mathbf{1}) - \alpha_0(T; \mathbf{0})). \tag{6}$$

Here we discuss how standard methods would estimate (6). Our goal is to illustrate the fundamental assumptions underpinning each method, and compare with our Assumptions 2 and 3. To illustrate we will assume a specific value $c = (0, 1, 0, 2, 0, 0, 0, 0, 1, 1)^{\mathsf{T}}$. In discussing these methods, we will mostly be concerned with how point estimates compare to the true value of the estimand, which here is $\tau = \$0.054$ using the experimental data in Table 2.

The naive approach would be to consider only the latest observed time point ($t_0 = 2$) under the experiment assignment $Z$, and use the observed population actions under $Z$ as an estimate for $\tau$; i.e.,

$$\hat{\tau}^{naive} = c^{\mathsf{T}}(\alpha_1(t_0; Z) - \alpha_0(t_0; Z)) = -\$0.051. \tag{7}$$

But this estimate to be unbiased for $\tau$, we generally require that

$$\alpha_1(t_0; Z) - \alpha_0(t_0; Z) = \alpha_1(T; \mathbf{1}) - \alpha_0(T; \mathbf{0}).$$

The naive estimate therefore makes a direct extrapolation from $t = t_0$ to $t = T$ and from the observed assignment $Z$ to the counterfactual assignments $Z = \mathbf{1}$ and $Z = \mathbf{0}$. This ignores, among other things, the dynamic nature of agent actions.

A more sophisticated approach is to analyze the agent actions as a time series. For example, Brodersen et. al. [6] developed a method to estimate the effects of ad campaigns on website visits. Their method was based on the idea of "synthetic controls", i.e., they created a time-series using different sources of information that would act as the counterfactual to the observed time-series after the intervention. However, their problem is macroeconometric and they work with observational data. Thus, there is neither experimental randomized assignment to games, nor strategic interference between agents, nor dynamic agent actions. More crucially, they do not study long-term equilbrium effects. By construction, in our problem we can leverage behavioral game theory to make more informed predictions of counterfactuals to time points after the intervention at which the distribution of outcomes has stabilized.

Another approach, common in econometrics, is the *difference-in-differences* (DID) estimator [7, 10, 16]. In our case, this method is not perfectly applicable because there are no observations before the intervention, but we can still entertain the idea by considering period $t = 1$ as the pre-intervention period. The DID estimator compares the difference in outcomes before and after the intervention for both the treated and control groups. In our application, this estimator takes the value

$$\hat{\tau}^{did} = \underbrace{c^{\mathsf{T}}(\alpha_1(t_0; Z) - \alpha_1(1; Z))}_{\text{change in revenue for game 2}} - \underbrace{c^{\mathsf{T}}(\alpha_0(t_0; Z) - \alpha_0(1; Z))}_{\text{change in revenue for game 1}} = -\$0.164. \tag{8}$$

This estimate is also far from the true value similar to the naive estimate. The DID estimator is unbiased for $\tau$ only if there is an additive structure in the actions [1], [3] (Section 5.2), e.g., $\alpha_j(t;Z) = \mu_j + \lambda_t + \epsilon_{jt}$, where $\mu_j$ is a policy-specific parameter, $\lambda_t$ is a temporal parameter, and $\epsilon$ is noise. The DID estimator thus captures a linear trend in the data by assuming a common parameter for both treatment arms ($\lambda_t$) that is canceled out in subtraction in Eq. (8). The extent to which an additivity assumption is reasonable depends on the application, however, by definition, it implies ignorability of the assignment (i.e., $Z$ does not appear in the model of $a_j(t;Z)$), and thus it relies on assumptions that are stronger than our assumptions [1, 3].

In a structural approach, Athey et. al. [4] studied the effects of timber auction format (ascending versus sealed bid) on competition for timber tracts. They estimated bidder valuations from observed data in one auction and imputed counterfactual bid distributions in the other auction, under the assumption of equilibrium play in both auctions. This approach makes two critical implicit assumptions that together are stronger than Assumption 3. First, the bidder valuation distribution is assumed to be a *primitive* that can be used to impute counterfactuals in other treatment assignments. In other words, the assignment is independent of bidder values, and thus it is strongly ignorable. Second, although imputation is performed for potential outcomes in equilibrium, which captures the notion of long-term effects, inference is performed under the assumption of equilibrium play in the *observed* outcomes, and thus temporal dynamic behavior is assumed away.

Finally, another popular approach to causality is through *directed acyclic graphs* (DAGs) between the variables of interest [17]. For example, Bottou et. al. [5] studied the causal effects of the machine learning algorithm that scores online ads in the Bing search engine on the search engine revenue. Their approach was to create a full DAG of the system including variables such as queries, bids, and prices, and made a Causal Markov assumption for the DAG. This allows to predict counterfactuals for the revenue under manipulations of the scoring algorithm, using only observed data generated from the assumed DAG. However, a key assumption of the DAG approach is that the underlying structural equation model is stable under the treatment assignment, and only edges coming from parents of the manipulated variable need to be removed; as before, assignment is considered strongly ignorable. As pointed out by Dash [9] this might be implausible in equilibrium systems. Consider, for example, a system where $X \to Y \leftarrow Z$, and a manipulation that sets the distribution of $Y$ independently of $X, Z$. Then after manipulation the two edges will need to be removed. However, if in an equilibrium it is required that $Y \approx XZ$, then the two arrows should be reversed after the manipulation. Proper causal inference in equilibrium systems through causal graphs remains an open area without a well-established methodology [8].

Finally we note that there exists the concept of Granger causality [11], which remains important in econometrics. The central idea in Granger causality is predictability, in particular the ability of lagged iterates of a time series $x(t)$ to predict future values of the outcome of interest, which in our case is the population action $\alpha_j(t;Z)$. This causality concept does not take into account the randomization from the experimental design, which is key in statistical causal inference.

# 4    Application: Experiment of Rapoport and Boebel [18]

The following tables report the payoff matrix structure (Table 1 used by Rapoport and Boebel and the observed data (Table 2), as reported by McKelvey and Palfrey [15].

Table 1: Normal-form game in the experiment of Rapoport and Boebel (values $L$ and $W$ are specified as described in the body of the paper) [18].

|       | $a'_1$ | $a'_2$ | $a'_3$ | $a'_4$ | $a'_5$ |
|-------|--------|--------|--------|--------|--------|
| $a_1$ | W | L | L | L | L |
| $a_2$ | L | L | W | W | W |
| $a_3$ | L | W | L | L | W |
| $a_4$ | L | W | L | W | L |
| $a_5$ | L | W | W | L | L |

Table 2: Experimental data of Rapoport and Boebel [18], as reported by McKelvey and Palfrey [15]. The data includes frequency of actions for the row agent and the column agent in the experiment, broken down by game and session. Gray color indicates that we assume the data to be long-term and thus we hold them out of data analysis and only use them to measure predictive performance. *(Note: There are five total actions available to every player according to the payoff structure in Table 1. The frequencies for actions $a_5, a'_5$ can be inferred because $\sum_{i=1}^{5} a_i = 1$ and $\sum_{i=1}^{5} a'_i = 1$.)*

|      |        | row agent | | | | column agent | | | |
|------|--------|-------|-------|-------|-------|--------|--------|--------|--------|
| Game | Period | $a_1$ | $a_2$ | $a_3$ | $a_4$ | $a'_1$ | $a'_2$ | $a'_3$ | $a'_4$ |
| 1 | 1 | 0.308 | 0.307 | 0.113 | 0.120 | 0.350 | 0.218 | 0.202 | 0.092 |
| 1 | 2 | 0.293 | 0.272 | 0.162 | 0.100 | 0.333 | 0.177 | 0.190 | 01.40 |
| 1 | 3 | 0.273 | 0.350 | 0.103 | 0.123 | 0.353 | 0.133 | 0.258 | 0.102 |
| 1 | 4 | 0.295 | 0.292 | 0.113 | 0.135 | 0.372 | 0.192 | 0.222 | 0.063 |
| 2 | 1 | 0.258 | 0.367 | 0.105 | 0.143 | 0.332 | 0.115 | 0.245 | 0.140 |
| 2 | 2 | 0.290 | 0.347 | 0.118 | 0.110 | 0.355 | 0.198 | 0.208 | 0.108 |
| 2 | 3 | 0.355 | 0.313 | 0.082 | 0.100 | 0.355 | 0.215 | 0.187 | 0.110 |
| 2 | 4 | 0.323 | 0.270 | 0.093 | 0.105 | 0.343 | 0.243 | 0.168 | 0.107 |

# 5    More details on Bayesian computation

Here we offer more details about the choices in implementing Algorithm 1 in Section 4.1 of the main paper. For convenience we repeat the content of Section 4.1 in the main paper and then expand with our details.

**Step 1: Model parameters.** For simplicity we assume that the models in the two games share common parameters, and thus $(\phi_1, \psi_1, \lambda_1) = (\phi_0, \psi_0, \lambda_0) \equiv (\phi, \psi, \lambda)$, where $\lambda$ are the parameters of the behavioral model to be described in Step 8. Having common parameters also acts as regularization and thus helps estimation. We emphasize that this simplification is not necessary as we could have two different set of parameters for each game. It is crucial, however, that the parameters are stable with respect to the treatment assignment because we need to extrapolate from the observed assignment to the counterfactual ones.

**Step 4: Sampling parameters and initial behaviors** As explained later we assume that there are 3 different behaviors and thus $\phi, \psi, \lambda$ are vectors with 3 components. Let $x \sim U(m, M)$ denote

that every component of $x$ is uniform on $(m, M)$, independently. We choose diffuse priors for our parameters, specifically, $\phi \sim \mathrm{U}(0, 10)$, $\psi \sim \mathrm{U}(-5, 5)$, and $\lambda \sim \mathrm{U}(-10, 10)$. Given $\phi$ we sample the initial behaviors in the two games as $\beta_1(0; Z) \sim \mathrm{Dir}(\phi)$ and $\beta_0(0; Z) \sim \mathrm{Dir}(\phi)$, independently.

Regarding the particular choices of these distributions, we first note that $\phi$ needs to have positive components because it is used as an argument to the Dirichlet distribution. Larger values than 10 could be used for the components of $\phi$ but the implied Dirichlet distributions would not differ significantly than the ones we use in our experiments. Regarding $\lambda$ we note that its components are used in quantities of the form $e^{\lambda[k]u}$ and so it is reasonable to bound them, and the interval $[-5, 5]$ is diffuse enough given the values of $u$ implied by the payoff matrix in Table 1. Finally the prior for the temporal model parameters, $\psi$, is also diffuse enough. An alternative would be to use a multivariate normal distribution as the prior for $\psi$ but this would not alter the procedure significantly.

**Steps 5 & 7: Pivot to counterfactuals.** Since we have a completely randomized experiment (A/B test) it holds that $\rho_Z = 0.5$ and therefore $\beta^{(0)} = 0.5(\beta_1(0; Z) + \beta_0(0; Z))$. Now we can pivot to the counterfactual population behaviors under $Z = \mathbf{1}$ and $Z = \mathbf{0}$ by setting $\beta_1(0; \mathbf{1}) = \beta_0(0; \mathbf{0}) = \beta^{(0)}$.

**Step 8: Sample counterfactual behavioral history.** As the temporal model, we adopt the *lag-one vector autoregressive model*, also known as VAR(1). We transform[1] the population behavior into a new variable $w_t = \mathrm{logit}(\beta_1(t; \mathbf{1})) \in \mathbb{R}^2$ (also do so for $\beta_0(t; \mathbf{0})$). Such transformation with a unique inverse is necessary because population behaviors are constrained on the simplex, and thus form so-called compositional data [2, 12]. The VAR(1) model implies that

$$w_t = \psi[1]\mathbf{1} + \psi[2]w_{t-1} + \psi[3]\epsilon_t, \tag{9}$$

where $\psi[k]$ is the $k$th component of $\psi$ and $\epsilon_t \sim \mathcal{N}(0, I)$ is i.i.d. standard bivariate normal. Eq. (9) is used to sample the behavioral history, $B_j$, from $t = 0$ to $t = T$, as described in Step 8 of Algorithm 1.

Such sampling is straightforward to do. We simply need to sample the random noises $\epsilon_t$ for every $t \in \{0, \ldots, T\}$, and then compute each $w_t$ successively. Given the sample $\{w_t : t = 0, \ldots, T\}$ we can then transform back to calculate the population behaviors $\beta_1(t; \mathbf{1}) = \{\mathrm{logit}^{-1}(w_t) : t = 0, \ldots, T\}$—for $B_0$ we repeat the same procedure with a new sample of $\epsilon_t$ since the two games share the same temporal model parameters $\psi$.

**Step 9: Behavioral model.** Here we rewrite the specifics of the behavioral model with more details. In $\mathrm{QL}_p$ agents possess increasing levels of sophistication. Following earlier work [19], we adopt $p = 3$, and thus consider a behavioral space with three different behaviors $\mathcal{B} = \{b_0, b_1, b_2\}$.

Recall that a behavior $\in \mathcal{B}$ represents the distribution of actions that an agent will play conditional on adopting that behavior. In $\mathrm{QL}_p$ such distributions depend on an assumption of *quantal response*, which is defined as follows. Let $u \in \mathbb{R}^{|\mathcal{A}|}$ denote a vector such that $u_a$ is the expected utility of an agent taking action $a \in \mathcal{A}$, and let $G_j$ denote the payoff matrix in game $j$ as in Table 1. If an agent is facing another agent with strategy (distribution over actions) $b$, then $u = G_j b$. The quantal best-response with parameter $x$ determines the distribution of actions that the agent will take facing expected utilities $u$, and is defined as

$$\mathrm{QBR}(u; x) = \mathrm{expit}(xu), \tag{10}$$

where, for a vector $y$ with elements $y_i$, $\mathrm{expit}(y)$ is a vector with elements $\exp(y_i)/\sum_i \exp(y_i)$. The parameter $x \geq 0$ is called the *precision* of the quantal best-response. If $x$ is very large then the response is closer to the classical Nash best-response, whereas if $x = 0$ the agent ignores the utilities and randomizes among actions.

Let $\lambda = (\lambda[1], \lambda[2], \lambda[3])$ be the precision parameters. Let $\alpha(b)$ denote the distribution over actions implied for an agent who adopts behavior. Given $\lambda$ the model $\mathrm{QL}_3$ calculates $\alpha(b_k)$, for $k = 0, 1, 2$, as follows:

- Agents who adopt $b_0$, termed *level-0* agents, have precision $\lambda_0 = 0$, and thus will randomly pick one action from the action space $\mathcal{A}$. Thus,

$$\alpha(b_0) = \mathrm{QBR}(u; 0) = (1/|\mathcal{A}|)\mathbf{1},$$

  regardless of the argument $u$.

- An agent who adopts $b_1$, termed *level-1* agent, has precision $\lambda[1]$ and assumes that is playing against a level-0 type agent. Thus, the agent is facing a vector of utilities $u_1 = G_j b_0$, and so

$$\alpha(b_1) = \mathrm{QBR}(u_1; \lambda[1]).$$

- An agent who adopts $b_2$, termed *level-2* agent, has precision $\lambda[3]$ and assumes is playing against a level-1 agent with precision $\lambda[2]$. Thus, it estimates that it is facing strategy $\alpha_{(1)2} = \mathrm{QBR}(u_1; \lambda[2])$, where $u_1 = G_j b_0$ as above. The expected utility vector of the level-2 agent is $u_2 = G_j \alpha_{(1)2}$, and thus

$$\alpha(b_2) = \mathrm{QBR}(u_2; \lambda[3]).$$

Given $G_j$ and $\lambda$ we can therefore write down a $5 \times 3$ matrix $Q_j = [\alpha(b_0), \alpha(b_1), \alpha(b_2)]$ where the $k$th column is the distribution over actions played by an agent conditional on adopting behavior $b_{k-1}$.

Conditional on population action $\beta_j(t; Z)$ the expected population action is $\bar{\alpha}_j(t; Z) = Q_j \beta_j(t; Z)$. The population action $\alpha_j(t; Z)$ is distributed as a multinomial with expectation $\bar{\alpha}_j(t; Z)$, and so $P(\alpha_j(t; \mathbf{1})|\beta_j(t; \mathbf{1}), G_j) = \mathrm{Multi}(|\mathcal{I}| \cdot \alpha_j(t; \mathbf{1}); \bar{\alpha}_j(t; \mathbf{1}))$, where $\mathrm{Multi}(n, p)$ is the multinomial density of observations $n = (n_1, \ldots, n_K)$ with expected frequencies $p = (p_1, \ldots, p_K)$. Hence, the full likelihood for observed actions in game $j$ required in Steps 10 and 11 of Algorithm 1 is given by the product

$$P(\mathcal{D}_j | B_j, \lambda_j, G_j) = \prod_{t=0}^{T-1} \mathrm{Multi}(|\mathcal{I}| \cdot \alpha_j(t; j\mathbf{1}); \bar{\alpha}_j(t; j\mathbf{1})).$$

## Footnotes

[1]The map $y = \mathrm{logit}(x)$ is defined as the function $\Delta^m \to \mathbb{R}^{m-1}$ such that, for vectors $y = (y_1, \ldots, y_{m-1})$ and $x = (x_1, \ldots, x_m)$, $\sum_i x_i = 1$, and $x_1 \neq 0$ wlog, indicates that $y_i = \log(x_{i+1}/x_1)$, for $i = 1, \ldots, n - 1$.