[Reviews · NeurIPS 2016]

Reviewer 1

Summary

This paper addresses an issue that is common in randomized controlled trials (or A/B tests): inferences may be invalid if there is nonstationarity, and in particular if users strategically respond dynamically to the differences in the two cells. The authors attempt to calibrate a strategic model using short-run data to then estimate a long-run causal effect.

Qualitative Assessment

---The way you refer to "causal effect" and "the fundamental problem of causal inference" is somewhat nonstandard. Typically (in the Rubin potential outcomes model, which is what you are building on), the causal effect is defined at the individual level, with a "treatment" outcome and "control" outcome for each experimental unit. The fundamental problem of causal inference is that only one of these two outcomes is actually observed for each experimental unit. You seem to be focusing on a slightly different issue, which is that the effect of treating the entire population cannot be determined correctly from just data when half the population is treated. It seems to me that this issue -- which can arise due to a variety of violations of the SUTVA assumption -- can exist independent of whether there is a multiagent interaction. Conversely, it seems multiagent considerations are relevant even when defining causal effects at the sub-population level. ---Your references are somewhat limited to recent work. There has been extensive work on causal inference in nonstationary environments, particularly by econometricians and time series analysts. It would be good to situate your work better relative to this work. Your main difference lies in the particular structure of behavioral model you assume, and I judged the paper on this basis. ----A significant limitation that I see is that the modeling approach requires games and behaviors to be defined before the estimation procedure can be carried out. One could, instead, imagine a fully nonparametric approach to causal inference that just aimed to remove any nonstationarity in treatment and control before estimating a treatment effect. The approach you take requires that one should believe constructing the necessary spaces of games and behaviors would be both reasonably specificed, and computationally tractable. ---The main contribution is in the modeling, as the theorem follows naturally from the modeling assumptions. The theorem is making an asymptotic statement imprecisely; it would be better to formally state the result as well. ---The numerics are carried out for a relatively simplified two player game, whereas the motivating examples are all drawn from ad auctions. I would like to understand whether this approach can be reasonably used in a more practical setting. For example, in an ad auction setting, how would you restrict the behaviors that you consider? As the paper currently stands, the example is a bit too simple, considering the significant model complexity that would arise if one tried to capture all behaviors possible in more complex (and real-world) strategic environments. ---Overall, I thought the authors took an intriguing approach to a very difficult problem. I was generally left with the impression that the work is promising but still preliminary, and would benefit from some deeper investigation of the applicability of the methods, as well as comparison to the very extensive related literature.

Confidence in this Review

2-Confident (read it all; understood it all reasonably well)


Reviewer 2

Summary

This paper pursues an interesting line of research: how to identifying the long term causal impact in randomized experiments where agents' react to experiment and adapt with time? Much of the literature on causal inference focus on generalization of inference made from small sample to the entire population. The authors pursue a novel line of research by focusing on long term effect of experiments. They do so by building a dynamic model for behavior update, fitting this model on short term data, and using the model to make long term claim.

Qualitative Assessment

I like the novelty the authors bring by focusing on an important problem that hasn't been studied well, and in using tools from behavioral game theory and machine learning. My concerns are mostly around generalization of this framework for real life scenarios and its robustness esp. around the dynamic update model for behavior, as well as action given behavior. Most of these seem to put a heavy burden of knowing the exact payoff, fixed agents, and working with finite games. I, however, am hoping that some of these would be studied more in their future work.

Confidence in this Review

1-Less confident (might not have understood significant parts)


Reviewer 3

Summary

The paper proposes a methodology to estimate the long-term causal effect of a treatment (such as a price increase) in a complex dynamical multiagent systems. To do so, it has to solve two inferential tasks: infer outcomes across assigments to the control or the treatment; infer across time. This is particularly challenging as only short-term experimental data are observed.

Qualitative Assessment

The paper is well written, the supplementary material exhaustive. Unfortunately, I am not qualified to give an educated opinion on its content. I am not familiar with many of the concepts used. This paper deserves a specialist in causality since its contribution may be significant. Naively, I wonder about the stability of such a methodology in practice (e.g. http://www.auai.org/uai2016/proceedings/papers/214.pdf). I will be amazed that such inference will yield robust conclusions with respect to real-life situations as in some fields (financial applications for risk and portfolios) estimating a simple yet robust variance-covariance is an endless problem.

Confidence in this Review

1-Less confident (might not have understood significant parts)


Reviewer 4

Summary

The paper deals with the long-term causal effects in a system where agents exhibit dynamic behaviours modeled under some assumptions. The key extension of the classical theory in the paper is the modeling of agent behaviors that can predict their actions based on game theory.

Qualitative Assessment

The paper is well written and well organized, there are some room for improvement: 1. The author defines the behavioral model and temporal model in a general form. It'd be interesting to see discussions on how different choice of models might influence the long-term effects. 2. As the counterfactuals always exist in real dataset, it might be relevant o design a simulated experiment and compare LACE, DID, etc.

Confidence in this Review

1-Less confident (might not have understood significant parts)


Reviewer 5

Summary

The authors consider the problem of understanding long-term causal effects of advertising interventions by brining ideas from game theory to bear on experimental data. Essentially, the authors use a deterministic (behavioral) game-theoretic model to extrapolate from short term inferences to the desired long-term ones. Similarly, they are able to look at interventions on an agent-by-agent bases, using a notion of inferred player types.

Qualitative Assessment

I think this is an interesting paper. I have two suggestions/criticisms. First, I believe that the algorithm describes what is essentially a two step procedure. The data is used to estimate a parameter, and then a model is used to convert that estimate into a different estimate. So it is effectively a plug-in estimator: to estimate g(theta) we first estimate theta and then use g(\hat{\theta}) as our estimate. In the present case g is a pretty elaborate dynamic model, but the idea is the same. If indeed this is the procedure I think an explanation could be much abbreviated and clearer as a result. Fortunately, the assumptions posited by the authors mean than the form of g() is not affected by the intervention or subsequent data, which allows for this two step procedure. Second, I think the authors are fairly glib/optimistic about the accuracy of behavioral game theory models. Given that the procedure relies on that estimate being spot-on, this is no small matter. Perhaps a citation to the more pessimistic assessment of Hahn, Mela and Goswami (2015) AoAS is warranted. The authors did an especially nice job at contrasting their approach to previous work (lines 76-82) and I find their approach to be very sensible in this light. Regarding the validity of the behavioral model, one can always perform sensitivity analyses for different choices of g().

Confidence in this Review

3-Expert (read the paper in detail, know the area, quite certain of my opinion)